# Free Flaps for Skin and Soft Tissue Reconstruction in the Elderly Patient: Indication or Contraindication

**DOI:** 10.3390/medsci11010012

**Published:** 2023-01-21

**Authors:** Heiko Sorg, Christian G. G. Sorg, Daniel J. Tilkorn, Simon Thönnes, Rees Karimo, Jörg Hauser

**Affiliations:** 1Department of Plastic and Reconstructive Surgery, Marien Hospital Witten, Marienplatz 2, 58452 Witten, Germany; 2Department of Health, University of Witten/Herdecke, Alfred-Herrhausen-Str. 50, 58455 Witten, Germany; 3Department of Management and Entrepreneurship, Faculty of Management, Economics and Society, University of Witten/Herdecke, 58455 Witten, Germany; 4Department of Plastic, Reconstructive and Aesthetic Surgery, Hand Surgery, Alfried Krupp Krankenhaus, Hellweg 100, 45276 Essen, Germany

**Keywords:** free tissue transfer, survival rate, radiation, tumor, fracture, smoker

## Abstract

Background: Increased lifespan and the improvement of medical treatment have given rise to research in reconstructive procedures in elderly patients. Higher postoperative complication rates, longer rehabilitation, and surgical difficulties remain a problem in the elderly. We asked whether a free flap in elderly patients is an indication or a contraindication and performed a retrospective, monocentric study. Methods: Patients were divided into two groups (YOUNG 0–59 years; OLD > 60 years). The endpoint was the survival of flaps and their dependence on patient- and surgery-specific parameters using multivariate analysis. Results: A total of 110 patients (OLD *n* = 59) underwent 129 flaps. The chance of flap loss increased as soon as two flaps were performed in one surgery. Anterior lateral thigh flaps had the highest chance for flap survival. Compared with the lower extremity, the head/neck/trunk group had a significantly increased chance of flap loss. There was a significant increase in the odds of flap loss in linear relation to the administration of erythrocyte concentrates. Conclusion: The results confirm that free flap surgery can be indicated as a safe method for the elderly. Perioperative parameters such as two flaps in one surgery and transfusion regimens must be considered as risk factors for flap loss.

## 1. Introduction

In recent decades, the total life expectancy at birth for men and women worldwide has increased to 72 years. This resulted in the highest-ever increase of 5.5 years between 2000 and 2016 since the 1960s. In Europe, life expectancy is even higher, averaging 77.5 years [1]. The global increase in median lifespan and improvement in medical treatment inevitably also result in new research questions, particularly in relation to surgical approaches to older patients [2,3]. Consecutively, this also leads to an increase in old and very old patients in plastic and reconstructive surgery. This patient population increasingly suffers skin and soft tissue injuries with considerable defect wounds on the extremities as well as in the head, neck, and trunk area due to increased risks caused by falls and complex diseases of an inflammatory or neoplastic origin. These lesions are additionally paired with a significant profile of secondary and preexisting diseases [4,5].

This often poses a special challenge to the reconstructive surgeon to ensure limb preservation on the one hand but also to restore the quality of life for the elderly patient on the other hand. Some groups have already shown that the incidence of complications of free tissue transfer is not necessarily related to the age but rather to the preoperative health status of the patient and that free tissue transfer can be performed as successfully in the elderly as in the younger patient [6,7]. In this case, however, age and flap survival cannot be the only criteria to be considered, but other parameters have to be taken into account in order to provide a valid conclusion. Despite the encouraging evidence to this point, there is still a high level of concern in practice about surgical intervention for old or very old patients, as caregivers are asked to perform surgery on patients who would not have been candidates for surgery in the past [2]. This is based on both the higher postoperative complication rate and longer rehabilitation time, as well as concerns about surgical difficulties in older patients [3,7]. Furthermore, decisions are made with little evidence, mostly derived from studies in which older age was a limiting factor for recruitment. Due to a specific focus on geriatrics and age-related traumatology in our hospital, significantly more patients in this age group are presented to our department of plastic surgery. We, therefore, asked ourselves, based on our own patient population, whether free flap reconstruction for skin and soft tissue defects in elderly patients is an indication or should rather be considered a contraindication. We also wanted to examine whether there are parameters to be considered in the preoperative preparation or in the perioperative setting of free flap reconstruction that could serve as parameters for improving flap survival.

## 2. Materials and Methods

All free flap plastic surgeries used for reconstruction of the skin or the soft tissue and performed in our department (monocentric) between January 2016 and March 2019 were recorded. Bone reconstructions using free flaps have been excluded from this study. Patients were divided into two descriptive groups to show the possible influence of age: YOUNG 0 to 59 years and OLD ≥ 60 years. This retrospective study was based on our own patient samples, using standardized and ethically validated research questions and methods. For the definition of age, we used the classification of the World Health Organization and other works in this field [8,9,10,11,12]. These work with the following definition of ages: transition to old age: 60- to 65-year-olds, young old: 60- to 74-year-olds, and aged and very old: 75- to 89-year-olds. As we could not find any significant further differences within the OLD group, this has been left, and only OLD vs. YOUNG groups to describe the cohorts are presented. Therefore, we have classified our patient cohort accordingly.

For statistical analyses, age was used as a metric variable since better modeling was possible in contrast to the bivariate variable young age vs. old age. The survival of the flap surgery and the relation of this survival to other factors were investigated as endpoints. For this purpose, the following additional parameters were recorded for each patient. Patient-specific parameters such as gender, smoking, body mass index (BMI), co-morbidities, risk factors, and pre-treatment (radiation, chemotherapy) were collected. Furthermore, surgery-specific parameters such as the American Society of Anesthesiologists (ASA) classification, flap type, recipient region, reasons for free flap transfer, surgery duration, hospital stay, blood transfusions, and flap survival were assessed.

Statistical analysis was performed by multivariate analysis using the statistical program R (R version 3.5.1 (Feather Spray); R Core Team 2018; Vienna, Austria). The basic model for statistical analysis referred to survival from flap surgery as the main outcome. A logistic regression was calculated for binomial data (Package: Ime4, version 1.1-18-1; Innsbruck, Austria) [13], in which loss of free flap was the dependent variable and age (as a metric variable), as well as sex (as a nominal variable) were the independent variables as an integral part of all statistical models. Because multiple flaps were sometimes performed within one surgery and patients received multiple surgeries, both patients and surgeries were included as random effects. Model selection was calculated using the Akaike information criterion (AIC) within likelihood-based inference. Additional influence variables were tested sequentially within the base model. The significance level was assumed to be *p* < 0.05.

This study protocol was reviewed and approved by Hannover Medical School, approval number 2106-2013.

## 3. Results

A total of 111 patients underwent free flap surgery during the period. One patient died so soon that no reliable statement on the outcome was possible (dropout; remaining *n* = 110). Therefore, 110 patients were included in the study (Table 1), who were treated in 111 hospital stays, in 124 operations with a total of 129 flaps. The YOUNG group consisted of *n* = 51, the OLD group of *n* = 59 patients. Further description of the included cohort is described in Table 1, Table 2 and Table 3.

The survival rate of free flap surgery was the same in both groups, subdivided according to age (Table 3). Prior irradiation had no significant effect on flap loss (group without Rdx 10.2% flap loss; group Rdx 18.2% flap loss; *p* = 0.7563). This was not significant even when performing a Monte Carlo simulation with 2,000 replicates, which was calculated due to the small sample size (Pearson Square Test *p* = 0.5872). Subsequent analyses regarding influencing factors showed the following results: Once two free flap procedures were performed in one surgery, the chance of loss increased with an odds ratio (OR) of 7.6 (IC: 1.2–46.3; AIC 95.7; *p* = 0.0283).

Compared to the reference group of anterior lateral thigh flaps (ALT), other flaps had an OR of 5.0 (IC: 1.2–20.3; AIC 96.2; *p* = 0.0247), indicating a higher chance of flap loss. Compared to the lower extremities as the recipient region, the group of head/neck/trunk region as a recipient had a significantly increased OR of 4.6 (IC: 1.3–16.3; AIC 96.5; *p* = 0.0178) for flap loss. Likewise, there was a significant increase in the probability of flap loss in linear relation to perioperative administration of erythrocyte concentrates with an OR of 1.4 (IC 1.3–16.3; AIC 95.3; *p* = 0.0257).

In addition, the number of revision surgeries performed (*n* = 10/129) was significant (*p* < 0.001) but cannot be measured in isolation as an influencing factor in this computational model. Revision surgery was scored as perioperative flap revision due to perfusion problems of free flap reconstruction. All other tested variables (ASA, reason for surgery, BMI, smoking, duration of surgery, subsidiary flap, intraoperative administration of anticoagulants) did not significantly influence the survival of flap surgery.

## 4. Discussion

The aim of the study was to find out whether free flap transfer is an indication or a contraindication for elderly patients. This should be followed by possible predictive parameters in the perioperative setting, which could be used to improve flap survival. The main findings were: (1) age should not play a role in consideration of performing free flap surgery in patients who are eligible for it. (2) In our study, the ALT flap had the highest probability of flap survival when performing free flap surgery, while performing two flap surgeries in one patient and flap surgeries in the head/neck/trunk region had a significantly increased chance of flap loss. (3) We could see a significant increase in the odds ratios of flap loss in linear relation to the administration of erythrocyte concentrates.

At present, free microvascular tissue transfer is a powerful surgical method for the reconstruction of various complex skin and soft tissue defects. The loss of a free flap, however, continues to be considered a major complication, resulting in additional surgeries, hospitalizations, and costs. For this reason, safety and minimization of risk to the patient are paramount in the indication and surgical planning process. Consequently, it is important to know the reasons and factors that lead to an increased risk of flap failure or loss so that measures can be taken to reduce this risk. Many factors are described in the literature which may be associated with an impairment of the safety of free tissue transfer [14,15,16,17]. Free flap surgery is still often used with caution and restraint to cover large soft tissue defects in elderly patients. It is hypothesized that the increased number of concomitant diseases in older age and the long duration of such surgery could lead to increased complications. In addition, the division of patients into age groups is not always clear in the literature and sometimes makes direct comparisons with other studies very difficult. During the question of the effect of age on the performance of free flap surgery and possible complication parameters, the patients were divided into a group consisting of patients 60 years of age and older (OLD) and another group containing patients under 60 years of age (YOUNG). We, therefore, analyzed outcomes over a period of more than three years to identify risk factors for flap failure, with a focus on the elderly patient. Flap survival was determined to be the main marker of surgical outcome and was compared with other parameters. Simultaneous performance of multiple flap procedures in one surgery and recipient region head/neck/trunk versus lower extremity were identified as contributory factors to the likelihood of limited flap survival. This is also reflected in the results of other studies, in which patients with free flaps in the head and neck region had a slightly higher rate of flap loss compared to other locations [18,19,20,21,22,23].

In general, the patient population with an indication for free flap reconstruction in the head and neck region is predominantly older, mostly has underlying oncological diseases in this region, and is therefore also associated with a higher rate of comorbidities. In this context, pre- or postoperative radiation of the recipient region of the free flap is also more frequent, which in itself can be regarded as a risk factor, especially in the case of delayed reconstructions [19].

In addition, patients in the head and neck region are subjected to higher doses of radiation (median 64 Gray [19]). This might cause additional damage to the recipient region of a free flap as a whole and to the blood vessels, which might lead to complications. However, the radiotherapy in our as well as other studies is not directly associated with a higher rate of flap loss (up to 10% [19]), while the rate of flap revision in our patient cohort was highest in the head and neck region and with the use of the free radial flap, compared to all other locations and flap types [23]. As a potential pathophysiological mechanism, impaired fibrinolysis of irradiated microvessels has been described. Thus, and if possible, radiation is recommended after flap reconstruction, as well as the use of fibrinolytic drugs during revisions in irradiated areas [19]. In our study, further analysis of the data showed that the slightly higher flap loss rates in our group compared to other groups had the same causes. Flap loss in our group was highest in the head/neck/trunk group in particular.

In the context of performing free flap surgery, perioperative conditions such as anemia, preoperative hemoglobin level, intraoperative blood loss, and the related issue of blood transfusion play a crucial role in addition to the consideration of patient-specific features [24,25,26,27]. This issue is not clear on closer examination of the published data and is therefore controversial. In our study, a linear dependence of flap loss on perioperative administration of erythrocyte concentrates was found, which is more in line with recommendations for less aggressive transfusion and consequent reduction of associated intrinsic complications. Kim et al. 2017 even showed in their analysis that transfusion had a 3.6-fold increased risk of flap loss. Of interest, the authors nevertheless come to the recommendation of an unhesitating transfusion for free flap transfer [25]. In other studies, this was not shown as clearly but was associated with significantly higher adverse and especially medical complications (i.e., pneumonia, re-intubation and prolonged ventilation, postoperative transfusion) after free tissue transfer [27]. In addition, it was shown that the revision rate was significantly increased after intraoperative blood transfusion within a period of 30 days [27].

As a "workhorse" of plastic surgery, our study showed a significantly higher survival probability of ALT flaps compared to the group of other flaps (except latissimus dorsi muscle flaps [28]). Based on this, a recommendation can be made in the planning of defect coverage in old age in favor of this type of flap if it is technically feasible. However, with regard to the perioperative administration of blood products, the ALT flap appears to be more likely to be associated with higher transfusions in the context of reconstruction with free flap surgery in the head and neck region than other flap types (free radial flap, free fibula [29]). Interestingly, different reasons for possible flap loss are shown in different body areas. For example, a large retrospective study using multivariate analysis in breast reconstruction by free flap shows that flap type, postoperative bleeding, and perfusion disturbance were seen as independent parameters for flap loss [18]. In free flap surgery of the head and neck region, this involved anastomosis to the recipient vessel (superficial temporal artery) and postoperative perfusion disturbances [18]. In the lower extremity, the presence of diabetes mellitus and anesthesia time were identified as parameters for flap loss [18]. Age did not play a role in this context. In children, too, comorbidities play only a minor role in the question of possible flap loss [30]. In addition, the success of free flap surgery in children tends to be measured by the secondary surgeries required or the rehabilitation time of the children [30]. In fact, one study showed that flap survival actually increased with age [6].

Single testing of other patient characteristics such as BMI, smoking, or duration of surgery revealed no direct influence on the likelihood of free flap survival. This might also be a limitation of the study due to sample size; however, the chosen statistical method is a suitable tool for the corresponding assessment and calculation. Although in other studies, ASA status, as well as surgery duration, were shown to be significant predictors of postoperative complications in the elderly patient, this could not be demonstrated in our cohort [6,14,31]. This may be due to the very balanced patient groups with almost the same ASA status as well as the almost equal operation durations of the two groups studied. Howard et al. studied the complication rate of free flaps in patients over 70 years of age [31]. Patients were divided into two groups (Group I = 70–79-year-old patients; Group II = over 80-year-old patients). Although the flap survival rate was similarly good in both collectives (> 96.5%), however, the continuing medical complications were significantly higher in group II than in group I. The flap survival rate was higher in group I than in group II. Furthermore, the overall complication rate and perioperative mortality rate were also higher in group II. Thus, this study demonstrates that age, as an independent variable, was significantly associated with general and medical complications but not with surgical ones [31]. In contrast, another older study reached a different conclusion. In this study, patients over 65 years of age were compared with patients under 65 years of age with free flap surgery [32]. In the older patients, both the number of pre-existing conditions and the complication rate was higher. Medical complications were also more frequent in older patients, whereas the incidence of wound healing complications was the same in both groups. Interestingly, it was observed that there were no longer significant differences between the two groups after the presence of pre-existing conditions was resolved. Based on this, the authors of this study concluded that age alone was not a risk factor for free flap surgery as well [32].

The ASA score has also proven to be a valuable parameter for the most important determinant of postoperative complications after microvascular surgery in reconstructive free flaps. For this reason, we have also included this parameter in our study. Although this study was performed fifty years ago, it described the fact that geriatric patients are associated with increased mortality during major surgical interventions; this finding held true for a long time with regard to free microvascular tissue transfer [33]. Since then, however, not only the surgical procedures but also the techniques of anesthesia and monitoring of patients have improved so that free tissue transfer is now a routine operation in the armamentarium of plastic-reconstructive surgeons and can be performed at any age with a high success rate. Nevertheless, a meaningful risk assessment before surgery is an important task in order to be able to assess possible postoperative complications together with the patient and to make appropriate preparations. In this regard, Kulakli-Inceleme and colleagues have described that older age is not associated with higher rates of serious complications or flap failures. However, increased rates of comorbidities were found (i.e., hypertension, peripheral artery disease, diabetes mellitus, and obesity) [34]. The ASA score is now considered in most studies to be a standardized and reliable tool for assessing and analyzing surgical risk for decision-making in microvascular reconstruction. However, it is also a highly subjective assessment by the anesthesiologist, as several studies showed significantly weak inter-rater reliability [35]. Even when other parameters, such as the G8 score, are used to define frailty as a parameter for postoperative mortality, morbidity, and prolonged recovery, no significant differences could be demonstrated in the assessment of these values compared with the ASA score [36]. As shown in our study but also in other studies, free flap transfer per se is no longer a contraindication for elderly or very old patients. Thus, it can be stated that the ASA score does not seem to be an adequate parameter (anymore) to assess the risk of free flap surgery but may (only) be able to evaluate the postoperative complications of the patient and length of hospital stay (Freeman), but not the flap survival per se.

Yet, in the context of this study, the topic of frailty assessment should also be considered in the discussion. Although the two parameters, age, and frailty, are closely related, they have to be considered and discussed strictly separately. In our study, we investigated several possible parameters (e.g., age, BMI, smoking, ASA) influencing flap survival. Other studies have also evaluated frailty as another parameter for a possible perioperative risk stratification. It was found that the determination of frailty indices or scores may be a more helpful indicator for flap reconstruction, e.g., in the head and neck region or in breast reconstruction, than age itself [12,37,38,39]. Thus, higher frailty scores have been found to be associated with a negative outcome after free flap surgery [39], and that frail patients benefit from being cared for in the intensive care unit after flap surgery [12,40].

## 5. Conclusion

In summary, the results of the current study confirm that free flaps can be indicated, with the same survival rate in elderly patients as in young patients, as a safe method for defect coverage of complex skin and soft tissue conditions [19,27,41,42]. More important in this context seems to be the reduction of perioperative risks, such as radiotherapy of the surgical site and the transfusion regime and the consideration and treatment of secondary diseases or the usage of other indices to stratify the perioperative risks and outcomes.

## Figures and Tables

**Table 1 medsci-11-00012-t001:** Presentation of descriptive statistics of the included patient cohort and the division into the two groups (YOUNG, OLD). * = number of missing records.

Patient Data/Group	YOUNG (4–59)	OLD (≥60)	ALL
Number (*n*)	51	59	110
Female	25 (49.0%)	28 (47.5%)	53 (48.2%)
Age, mean (±SD)	43.9 (±14.9)	72.7 (±7.7)	59.3 (±18.5)
Smoker n (%)	21 (41.2%)	12 (20.7%) * 1	33 (30.3 %) * 1
BMI Group			
0 (BMI 15 to < 30)	43 (84.3%)	39 (68.4%) * 2	82 (75.9 %) * 2
1 (BMI 30 to < 40)	7 (13.7%)	16 (28.1%) * 2	23 (21.3 %) * 2
2 (BMI 40 and more)	1 (2.0%)	2 (3.5%) * 2	3 (2.8 %) * 2
ASA-Classification (1 to 6)			
1—normal	8 (16.0%) * 1	6 (10.3%) * 1	14 (13.0 %) * 2
2—mild general disease	31 (62.0%) * 1	26 (44.8%) * 1	57 (52.8 %) * 2
3 + 4—severe general disease	11 (22.0%) * 1	26 (44.8%) * 1	37 (34.3 %) * 2

* = number of missing records; SD, standard deviation; BMI, Body Mass Index; ASA, American Society of Anesthesiologists.

**Table 2 medsci-11-00012-t002:** Presentation of concomitant diseases of the studied cohort of patients and the division into the two groups (YOUNG, OLD).

Patient Data/Group	YOUNG (4–59)	OLD (≥60)	ALL
Tumor	16 (31.4%)	16 (27.1%)	32 (29.1%)
Previous cardiological disease	14 (27.5%)	46 (78.0%)	60 (54.6%)
Diabetes mellitus	4 (7.8%)	18 (30.5%)	22 (20.0%)
Thyroid disease	3 (5.9%)	14 (23.7%)	17 (15.5%)
Lung disease	6 (11.8%)	12 (20.3%)	18 (16.4%)
Liver disease	1 (2.0%)	3 (5.1%)	4 (3.6%)
Renal disease	3 (5.9%)	10 (17.0%)	13 (11.8%)
Neurological pre-existing disease	6 (11.2%)	14 (23.7%)	20 (18.2%)
Vascular disease	5 (9.8%)	15 (25.4%)	20 (18.2%)
Rheumatoid pre-existing disease	0 (0.0%)	4 (6.8%)	4 (3.6%)
Alcohol and drug abuse	3 (5.9%)	3 (5.1%)	6 (10.4%)
Chemotherapy	3 (5.9%)	2 (3.4%)	5 (8.8%)

**Table 3 medsci-11-00012-t003:** Presentation of the characteristics of the performed free flap plastic surgeries of the studied cohort of patients and the division into the two groups (YOUNG, OLD).

Flap Details/Group	YOUNG (4–59)	OLD (≥60)	ALL
Quantity	63	66	129
Flap type			
Anterior lateral thigh	27 (42.9%)	33 (50.0%)	60 (46.5%)
M. latissimus dorsi	13 (20.6%)	24 (36.4%)	37 (28.7%)
Other	23 (36.5%)	9 (13.6%)	32 (24.8%)
Recipient region			
Head/neck/trunk	19 (30.2%)	12 (18.2%)	31 (24.0%)
Upper extremities	10 (15.9%)	3 (4.5%)	13 (10.1%)
Lower extremities	34 (54.0%)	51 (77.3%)	85 (65.9%)
Reasons for free flap			
Fracture	18 (29.5%) * 2	17 (25.8%)	35 (27.6%) * 2
Tumor	19 (31.1%) * 2	14 (21.2%)	33 (26.0%) * 2
Infection	11 (18.0%) * 2	20 (30.3%)	31 (24.4%) * 2
Other	13 (21.3%) * 2	16 (24.2%)	29 (22.8%) * 2
Radiotherapy of the recipient region	6 (9.5%)	5 (7.6%)	11 (8.5%)
Additional skin graft	22 (34.9%)	25 (38.5%) * 1	47 (36.7%) * 1
Intraoperative revision	3 (4.8%) * 1	7 (10.6%)	10 (7.8%) * 1
Total flap loss	7 (11.1%)	7 (10.6%)	14 (10.9%)
Primary surgeries/group			
Number	59	65	124
Anticoagulant administration	33 (57.9%) * 2	39 (62.9%) * 3	72 (60.5%) * 5
Surgery duration in minutes, mean (±SD)	388.3 (122.5) * 1	385.2 (147.7)	386.7 (135.6) * 1
Number of erythrocyte concentrates given, mean (±SD)	0.3 (±0.9)	0.8 (±1.9)	0.6 (±1.5)
Hospital stay data/group			
Number of primary flap surgeries	59 (39.6%)	65 (38.0%)	124 (38.8%)
Number of flap revisions	8 (5.4%)	14 (8.2%)	22 (6.9%)
Stays with lethal end	2 (3.9%)	1 (1.7%)	3 (2.7%)
Length of stay in days, mean (±SD) **	30.0 (±21.8)	33.6 (±20.0)	32.0 (±20.8)

* n missing present with n records; ** only stays without a lethal outcome were included (*n* = 49/59/108); SD, standard deviation.

## Data Availability

Data are archived and can be obtained by the study team.

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
