# Peer review of "Free Flaps for Skin and Soft Tissue Reconstruction in the Elderly Patient: Indication or Contraindication"

_medsci, 2023, doi:10.3390/medsci11010012_

Round 1

Reviewer 1 Report

Thank you for submitting this study and shining a light on age and safety of flap surgery. In general, the study is well written and conceptualized.

Attached some minor comments

Abstract:

Ok, in general I like to include values in the results section, but this might only be personal preference

LL 16: Aim of the study before Methods.

Introduction

Ok. The aim of the study could have been presented more clearly. Maybe add a study hypothesis

M&M

Ok

LL 64 for = from ?

Results

Ok. Figures ok

Please add a detailed description of the flap revisions  

LL 91 treated in 111 stays means hospital stays?

LL 125: because underpowered for detailed subgroup analysis?

Discussion

Ok

Again, just a personal preference but I suggest to start the discussion like this:

Aim of the Study was ……

The main findings were (1) (2) (3)…

LL 163: remove of

LL 176- 186 This finding is interesting. Maybe add an explanation for the higher complication rate after blood transfusion (if there is one)

LL188 maybe define ALT flap for non-plastic surgeons

LL 189: Based on this, a recommendation can be made in the planning of defect 189 coverage in old age in favor of this type of flap if it is technically feasible.

Why? Is there an explanation?

LL 225 remove however

LL 240 remove just recently

LL 255 analysis presented here -> current study

Conclusion

Ok

Reviewer 2 Report

Reviewer comments

1.     Results. You mention the ALT group. Should this be the OLD group? Typo?

2.     The only real comment/question is that maybe you should have looked at making the cut-off age higher. Nowadays, most people who are 60 are not really considered “old” and lot of these patients can be quite healthy. Did you see what would happen if the age cut-off was for example 70? Would probably make more sense. Making the age cut-off 60 years is the biggest downside of this study.

Reviewer 3 Report

This study, attempt to define if there is an increase in morbidity in patients that are old. This has been well studied in the literature. Age does not seem to be an indicator of postoperative morbidity, or flap complications.  

 A major issue that needs to be addressed before this manuscript to be even considered for review is the definition of old. Most of the current literature stratify patience by age into perhaps 10 year for segments. Furthermore t 60 is not considered the cut off for Old in any series.

 The authors will need to review their data and rewrite the manuscript with an acceptable age definition.

Furthermore, the morbidity of reconstruction in the head and neck versus the extremities, etc. differs dramatically due to many factors. To lump all these cases together exposes not only a type two error with cofounding variables.

Round 2

Reviewer 2 Report

I am fine with the revisions

Author Response

Dear reviewer,

we are happy to read that you are fine with our revision. Thank you for this

All the best!

Reviewer 3 Report

Thank you for your comments. I find supporting your definition of old using a reference from more than 20 years ago is not helpful. The issue as you also state is not age it is frailty or whatever term you wish to use. The literature is full of much better papers that ask the same question and answer it.
